# Movement in low gravity environments (MoLo) programme–The MoLo-L.O.O.P. study protocol

Nolan Herssens[1]ᴼ*, James Cowburn[2,3]ᴼ, Kirsten Albracht[4,5,6], Bjoern Braunstein[4,5,7,8], Dario Cazzola[2,3], Steffi Colyer[2,3], Alberto E. Minetti[9], Gaspare Pavei[9], Jörn Rittweger[10,11], Tobias Weber[1,12], David A. Green[1,12,13]

1 Space Medicine Team, European Astronaut Centre, European Space Agency, Cologne, Germany, 2 Department for Health, University of Bath, Bath, United Kingdom, 3 Centre for the Analysis of Motion, Entertainment Research and Applications, University of Bath, Bath, United Kingdom, 4 Centre for Health and Integrative Physiology in Space, German Sport University, Cologne, Germany, 5 Institute of Movement and Neuroscience, German Sport University, Cologne, Germany, 6 Department of Medical Engineering and Technomathematics, University of Applied Sciences Aachen, Aachen, Germany, 7 Institute of Biomechanics and Orthopaedics, German Sport University, Cologne, Germany, 8 German Research Centre of Elite Sport Cologne, Cologne, Germany, 9 Laboratory of Physiomechanics of Locomotion, Department of Pathophysiology and Transplantation, University of Milan, Milan, Italy, 10 Division of Muscle and Bone Metabolism, Institute of Aerospace Medicine DLR, Cologne, Germany, 11 Department of Pediatrics and Adolescent Medicine, University of Cologne, Cologne, Germany, 12 KBR, Cologne, North Rhein-Westphalia, Germany, 13 Centre of Human and Applied Physiological Sciences, King's College London, London, United Kingdom

ᴼ These authors contributed equally to this work.
* nolan.herssens@esa.int

**Data Availability Statement:** All relevant data are within the paper and its Supporting Information files.

## Abstract

### Background

Exposure to prolonged periods in microgravity is associated with deconditioning of the musculoskeletal system due to chronic changes in mechanical stimulation. Given astronauts will operate on the Lunar surface for extended periods of time, it is critical to quantify both external (e.g., ground reaction forces) and internal (e.g., joint reaction forces) loads of relevant movements performed during Lunar missions. Such knowledge is key to predict musculoskeletal deconditioning and determine appropriate exercise countermeasures associated with extended exposure to hypogravity.

### Objectives

The aim of this paper is to define an experimental protocol and methodology suitable to estimate in high-fidelity hypogravity conditions the lower limb internal joint reaction forces. State-of-the-art movement kinetics, kinematics, muscle activation and muscle-tendon unit behaviour during locomotor and plyometric movements will be collected and used as inputs (Objective 1), with musculoskeletal modelling and an optimisation framework used to estimate lower limb internal joint loading (Objective 2).

**Funding:** This work was supported by the European Space Agency: ESA contract No. 4000123348/18/NL/MH with the University of Bath; ESA contract No. 4000133724/21/NL/AT with the University of Applied Sciences Aachen. The "Locomotion On Other Planets (L.O.O.P.): Hypogravity Analogue" of the University of Milan is supported by the European Space Agency as an ESA ground-based facility through the Continuously Open Research Announcement Opportunity for Ground-Based Facilities (ESA-CORA-GBF), ESA contract No. 4000137794/22/NL/PA/pt. This study was funded by CAMERA, the RCUK Centre of the Analysis of Motion, Entertainment Research and Applications, EP/M023281/1. Authors DG and TW are employed by KBR GmbH on behalf of the European Space Agency. The funder KBR GmbH provided support in the form of salaries for the authors DG and TW but did not have any role in the study design, data collection, and analysis, decision to publish, or preparation of the manuscript. The funders had and will not have a role in study design, data collection and analysis, decision to publish, or preparation of the manuscript.

**Competing interests:** The authors have declared that no competing interests exist.

## Methods

Twenty-six healthy participants will be recruited for this cross-sectional study. Participants will walk, skip and run, at speeds ranging between 0.56–3.6 m/s, and perform plyometric movement trials at each gravity level (1, 0.7, 0.5, 0.38, 0.27 and 0.16g) in a randomized order. Through the collection of state-of-the-art kinetics, kinematics, muscle activation and muscle-tendon behaviour, a musculoskeletal modelling framework will be used to estimate lower limb joint reaction forces via tracking simulations.

## Conclusion

The results of this study will provide first estimations of internal musculoskeletal loads associated with human movement performed in a range of hypogravity levels. Thus, our unique data will be a key step towards modelling the musculoskeletal deconditioning associated with long term habitation on the Lunar surface, and thereby aiding the design of Lunar exercise countermeasures and mitigation strategies.

## Introduction

Exposure to prolonged periods in microgravity (μg) is associated with multi-systems deconditioning [1, 2], including musculoskeletal [3–5]. Musculoskeletal deconditioning is evident by bone mineral density (BMD) and content (BMC) decrements, muscle atrophy, and loss of muscle strength [6], albeit with significant intra-individual variability [7]. Such de-conditioning presumably reflects adaptation driven metabolic factors [8], mechano-sensitivity at the level of the central nervous system [9] and musculoskeletal tissues themselves [10]. This sensitivity precipitates adaptation/remodelling that is 'appropriate' in μg, but functionally 'negative' in a gravitational environment. Such 'negative' remodelling occurs in muscle [11] and skeletal [12, 13] despite performance of daily resistance and aerobic exercise countermeasures on the International Space Station (ISS) [1].

Whilst it is planned for astronauts to return to the Lunar surface in the coming decade [14], and to stay for durations that will progressively be way in excess of those achieved in the Apollo era, the forces, and thus structural (musculoskeletal) requirements associated with locomotion (and potentially surface exercise) at hypogravity is unknown. Thus, it is critical to quantify both the external (e.g., ground reaction forces: GRFs) and internal (e.g., joint reaction forces and muscle-tendon forces) kinetics associated with human movement performed in Lunar (0.16g), and other gravity levels (g). Such knowledge is required not only to guide Lunar surface extravehicular activity (EVA) operations, but also to inform modelling of musculoskeletal deconditioning associated with long term habitation, and thereby potentially determine the need for, and definition of, Lunar exercise countermeasures.

The association between (simulated) hypogravity (0g < g < 1g) and the reduction of external kinetics has been investigated. For instance, studies conducted in the L.O.O.P. (Locomotion On Other Planets) Facility at the University of Milan demonstrated reduced external work when walking, running and skipping at speeds ranging between 0.56 and 3.6 m/s representing gait speeds observed in simulated hypogravity using vertical body weight support, compared to 1g [15, 16]. Furthermore, hip, knee and ankle net joint moments have been shown to decrease in proportion with simulated hypogravity during walking [17]. Similarly, previous work performed as part of the MoLo programme (MoLo VTF) reported peak vertical

GRFs to be scaled to simulated gravity levels (0.7, 0.38, 0.27 and 0.16g) during sub-maximal plyometric hopping on a vertical suspension system [18].

On the ISS and thus in µg, 'gravity replacement loading systems' are employed during exercise in an attempt to maintain external kinetics, including GRFs. Whilst comprehensive evaluation of external kinetics during exercise on the ISS has yet to be performed, 1g GRFs are not replicated on the T2 treadmill [19–21], or the Advance Resistive Exercise Device (ARED) [22] due to engineering and restraint system tolerability issues. The failure to replicate 1g equivalent external kinetics is also reflected in lower joint kinetics [17], and other indices of 'internal' musculoskeletal kinetics and thus 'loading', i.e., forces and moments acting on bones, joints, muscles and tendons [17, 23, 24]. However, due to the low number, methodological heterogeneity and potential bias of currently available studies, the relationship between movement in hypogravity and internal kinetics remains unknown [23].

Increasing evidence suggests that forces (loading, strain and strain rates) generated by muscular contraction may be important drivers for preserving musculoskeletal health via remodelling regulation, termed the muscle-bone hypothesis [25]. For instance, a reduction in forces acting through the Achilles Tendon may play a key role in the significant loss of BMD and BMC observed in the calcaneus [26] and altered neuromechanics of the calf muscle complex [27]. Recent evidence (acquired as part of the MoLo programme) suggests that gastrocnemius muscle-tendon unit (MTU) behaviour is preserved during walking [28], but not running [29] at 0.7g in a vertical suspension system. In contrast, hypogravity-induced modulation of joint kinematics and contractile behaviour differed between running in simulated Lunar and Martian gravities [28].

However, concurrent internal kinetics were not reported, which are critical to understand how hypogravity-induced reduction of external kinetics are associated with changes in internal joint forces. Estimates of the forces and moments experienced at joints, muscles, muscle-tendons, and at the bone can be derived from kinematic and kinetic data, via inverse dynamics [30]. Indeed, the relationship between external kinetics and internal forces is complex. As a result, estimation of external kinetics alone is insufficient to predict internal forces, and thus predict musculoskeletal adaptation via computational musculoskeletal (MSK) modelling [31].

MSK modelling is used to represent skeletal anatomy and muscle-tendon unit physiology, including geometry, contraction dynamics, and neural control, and characterise human movement [32]. When combined with optimisation techniques, musculoskeletal modelling is used to predict muscle activation patterns, and estimate internal kinetics, which is currently not directly measurable in vivo [32]. This approach has allowed researchers to distinguish between clinical populations [33, 34], and grade exercises according to their joint kinetic (loading) profile [35]. Yet, to our knowledge, only two hypogravity computational studies have been performed. The first evaluated a lower extremity assistive device for resistance training in microgravity [36]. The second predicted preferred locomotion strategies at low gravity levels and revealed three distinct locomotion strategies [37]. In 1g, walking was–unsurprisingly–predicted as the preferred locomotor strategy up to 2 m/s, while at Martian gravity, running was predicted to be most energy-efficient. In Lunar gravity however, skipping was predicted to be the optimal strategy–consistent with that frequently observed during Apollo missions [37–39]. However, neither of these studies reported muscle-tendon unit forces, nor joint reaction forces.

On Earth, the plyometric-like mechanics of skipping have been shown to result in greater vertical GRFs than running at the same speed [15, 37]. Although one study has shown that the peak forces borne by the knee are lower during skipping than during running at the same speed [40], which suggests that the proposed benefit of plyometric-like movements on musculoskeletal loading do not manifest at the joint level. However, McDonnell and colleagues [40]

used a musculoskeletal model that assumed muscles produce force proportional to physiological cross-sectional area, and do not account for neural input nor force-length-velocity relationships. Ignoring activation and contraction dynamics have been shown to produce non-physiological, instantaneous changes in force estimations during ballistic movements, which can lead to an over-reliance on muscles with large physiological cross-sectional area. To better understand the benefit of plyometric-like movements to hypogravity exercise countermeasures, it is important to achieve physiologically realistic muscle estimations.

Yet, in a recent long duration head down tilt bed rest (HDTBR) study, the pre-eminent ground-based analogue of μg [41], a short high-intensity jump training (i.e., 5–6 per week for ~4 mins) was demonstrated to be an efficient and effective exercise countermeasure ameliorating HDTBR-induced decrements in cardiovascular (e.g. $VO_2$max) [42] and musculoskeletal (e.g., loss of lean mass, bone mineral density and content) [42, 43]. Such findings are consistent with the generation of high external forces during plyometric contractions [44–46]. Furthermore, previous work performed as part of MoLo programme (MoLo VTF) demonstrated that peak GRFs during submaximal plyometric ankle hopping in simulated hypogravity (0.7, 0.38, 0.27, and 0.16g) were comparable to those generated during walking and running in 1g [18]. However, in this study, hop height was constrained by the experimental setup, and thus whether similar relationships are preserved during maximal hops and other forms of explosive maximal jumping (e.g., counter-movement jumping and drop-jumps) and the internal joint kinematics are unknown.

To our knowledge the only ground-based analogue that is able to allow evaluation of maximal jumping in simulated hypogravity is the L.O.O.P. facility in Milano, Italy. L.O.O.P. provides high fidelity hypogravity simulation via employment of very long (17m) twin (in series) calibrated (via electric winch) bungee cords, thereby minimising the relative change of bungee length, and hence recoil force during movement [47]. Furthermore, L.O.O.P. is equipped such that simultaneous 3D kinematics and GRFs can be assessed during locomotion and maximal plyometric movements. In recognition of this unique capability, ESA's Human Research Office recently incorporated L.O.O.P. into its ground-based facility (GBF) portfolio in order to facilitate access to the facility by external teams through its Continuously Open Research Announcement (CORA) programme.

Thus, the aim of this paper is to define an experimental protocol and methodology suitable to estimate in high-fidelity hypogravity conditions the lower limb internal joint reaction forces. State-of-the-art movement kinetics, kinematics, muscle activation and muscle-tendon unit behaviour during locomotor and plyometric movements will be used as inputs, with musculoskeletal modelling and an optimisation framework used to estimate lower limb internal joint loading.

### Objectives

The first objective of this study will be to collect state-of-the-art movement kinetics, kinematics, muscle activation and muscle-tendon unit data during locomotor and plyometric movements using a vertical weight support system simulating different hypogravity conditions (e.g., 1, 0.7, 0.5, 0.38, 0.27 and 0.16g).

The second objective, and main aim of the study will be to employ an optimal control framework with a full-body musculoskeletal model in order to estimate joint reaction forces of the ankle, knee and hip, using data-tracking simulations.

## Materials and methods

### Study design

This study will be of a cross-sectional study design. Participants will be asked to attend the L. O.O.P laboratory at the University of Milan on two occasions. The first visit will be used to

familiarise participants with the L.O.O.P. vertical body weight support system and the performance of walking, running, skipping and plyometric movement, and the second visit, for actual data collection (see *"Data Collection Procedure and Experimental Setup"*).

## Ethics approval

The study has received approval from the Research Ethics Approval Committee for Health of the University of Bath (ID: EP 18/19 018) and from the Ethical Board of the University of Milan (ID: 12/22). All participants will be asked to provide written informed consent prior to the study, in accordance with the Declaration of Helsinki, having received written and supplementary oral explanation in addition to having been given the opportunity to ask any questions they might have. All participants will be informed that they may withdraw from the study at any time up to the final collation of data, without being required to provide a reason.

## Study population

To be eligible for inclusion, volunteers can be of any gender but must fulfil the following eligibility criteria:

Inclusion:

- Healthy adults between the age of 18 and 64 years.

- Being physically active, defined as at least 30 minutes of moderate-to-vigorous physical activity three times per week.

- Individuals able to understand the explanations and instruction related to the present study, provided either in English or Italian.

Exclusion:

- Unable to walk, run, or jump independently without an assistive device.

- Any current lower-limb injury that prevents the participant from performing high impact movements.

- Any injury or condition that prevents the wearing of a safety harness, required for the body weight support system.

- Any medical condition or impairment that could potentially impede safe and comfortable participation.

**Recruitment.** All volunteer participants will be recruited through a combination of convenience and snowball sampling using advertisements across the University of Milan premises. Initial contact with current university staff and students and previous L.O.O.P. study volunteers will be made via word of mouth in line with the usual processes of the University of Milan.

**Sample size.** As to our knowledge there are no directly comparable published data, a sample size estimation was performed based on the results of a pilot study conducted at the L.O.O.P. GBF. During this pilot study, a single participant performed submaximal single-leg hopping at 2 Hz (using a metronome) for 30 seconds under five simulated gravity conditions (1, 0.5, 0.37, 0.25, 0.17g). For each session, peak vertical joint reaction forces were calculated for the hip, knee and ankle using the same simulation framework as described in this protocol (see *"Planned Data Analysis"*).

Using this simulation framework, means and standard deviations of the peak vertical joint forces of the hip, knee and ankle were calculated and extracted for each gravity condition, which were used to calculate the Cohen's *d* effect sizes based on the mean difference and pooled standard deviations [48]. These data were then used in an *a priori* power analysis to estimate a required sample size of 21 (S1 File) using G*Power (version 3.1.9) [49] with: t-test (matched pairs), $\alpha$ = 0.05, Power = 0.95, two-tailed. In order to ensure adequate data collection, we estimate a maximal data loss of 25%, therefore our sample size target will be 26 participants.

## Data collection procedure and experimental setup

Prior to participating in the Familiarisation Session and the Data Collection Session, and after having received written and supplementary oral explanation in addition to having been given the opportunity to ask any questions they might have, written informed consent of all volunteer participants will be collected.

**T0 –Familiarisation session.**   To familiarize with the L.O.O.P. vertical body weight support system, participants will initially perform a walking trial at 1.11 m/s, a skipping trial at 1.94 m/s and a running trial at 3.06 m/s, in addition to plyometric movements (fifteen submaximal hops, five submaximal countermovement jumps, three drop-jumps/landings) at each predefined simulated gravity level, in a non-randomized, descending sequence (1, 0.7, 0.5, 0.38, 0.27, 0.16g; Fig 1).

**T1 –Data collection session.**   After placing all motion capture markers and sensors on the participant's body (see *"Study Hardware"*), each participant will perform a short, pre-defined warm-up (five-minute walking at their preferred walking speed in 1g, fifteen submaximal ankle hops and five submaximal countermovement jumps in 1g).

Before the experimental dynamic trials, static motion capture calibration will be conducted with participants quietly standing on the instrumented treadmill centrally within the motion capture volume in a T-pose (arms outstretched horizontally) for five seconds. All participants will then be exposed to a randomised sequence of simulated gravities (1, 0.7, 0.5, 0.38, 0.27, 0.16g). For each gravity level, the conditions for the locomotion trials (i.e., walking, running and skipping) at various defined speeds (Table 1), and plyometric movement trials (i.e., maximal ankle hopping, maximum effort countermovement jumping, and drop-jumps/landings) will also be performed in a randomised order (Fig 1).

All locomotion trials (walking, running and skipping) at each gravity level will be performed for at least one minute in a randomized order at speeds previously shown (Table 1) to facilitate analysis of joint internal loads and muscle-tendon unit behaviour and in relation to the cost of transport curve as a function of speed and gravity [15, 16].

All plyometric movement trials will be performed in the gravity conditions in a randomized order:

- Vertical hopping will be performed in a single ramp-up-ramp-down trial consisting of 30 consecutive hops ascending from very shallow (<5cm) to maximal jump height, and then descending back to starting target height. Audio feedback (i.e., increasing and decreasing pitch) guiding hopping height will be provided similar to that used in our pilot study on the verticalized treadmill facility [18].

- Three maximal effort vertical countermovement jumps separated by at least 10 seconds.

- Three drop-jumps/landings separated by at least 10 seconds.

Self-selected rest periods will be encouraged between trials to minimise fatigue, and any discomfort associated with wearing the harness.

## $T_0$ –Familiarisation Session

*Simulated gravity levels (g)*

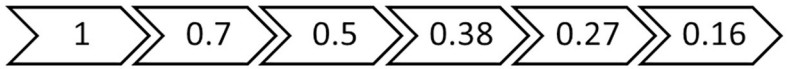

Performance of

*Locomotion Trials*
- Walking @ 1.11 m/s
- Skipping @ 1.94 m/s
- Running @ 3.06 m/s

*Plyometric Movements*
- 15 Submaximal ankle hops
- 5 Submaximal countermovement jumps
- 3 Drop-jumps/landings

## $T_1$ – Data Collection Session

*Warm-up @ 1g*
- Five-minute walking at preferred walking speed
- 15 Submaximal ankle hops
- 5 Submaximal countermovement jumps

*Simulated gravity levels (g) – Randomized*

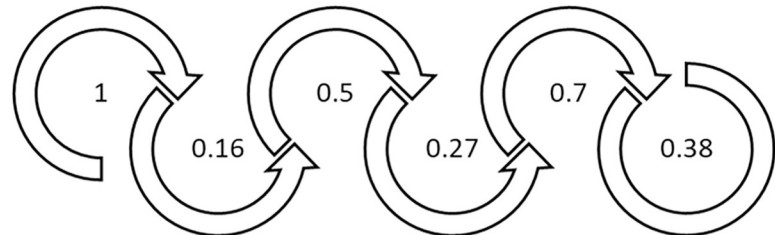

Performance of

*Locomotion Trials – Randomized*

|          | Speed (m/s) | | | | | | |
|----------|------|------|------|------|------|------|------|
|          | 0.56 | 1.11 | 1.39 | 1.94 | 2.50 | 3.06 | 3.61 |
| Walking  | X    | X    | X    |      |      |      |      |
| Skipping |      |      |      | X    | X    |      |      |
| Running  |      |      | X    | X    | X    | X    | X    |

- - - - - - - - - - - - - - - - - - - - - - ***Break*** - - - - - - - - - - - - - - - - - - - - - -

*Plyometric Movement Trials – Randomized*
- Vertical hopping: 30 consecutive hops from very shallow, to maximum jump height, to shallow again
- 3 maximum effort vertical countermovement jumps
- 3 drop-jumps/landings

**Fig 1. Schematic of the experimental setup and data collection procedure.**

**Table 1. Conditions of the locomotion trials.**

| Locomotion Style | Speed (m/s) | | | | | | |
|---|---|---|---|---|---|---|---|
| | 0.56 | 1.11 | 1.39 | 1.94 | 2.50 | 3.06 | 3.61 |
| Walking | X | X | X | | | | |
| Skipping | | | | X | X | | |
| Running | | | X | X | X | X | X |

**Safety assessment and risk prevention.** To mitigate risk of injury, in addition to experiencing a familiarisation protocol (Fig 1) participants will be required to wear their usual running/sport shoes. Risk of falling will also be mitigated by the unloading harness being attached to the pelvis and shoulders. Participants will also be instructed to report any discomfort experienced to allow mitigation via rest and/or harness adjustment.

**L.O.O.P. facility.** The L.O.O.P. facility is located within the cavaedium, a square (3 x 3 m) but tall (17 m) (Fig 2) space inside the Human Physiology building of the University of Milan where calibrated bungee cords provide body suspension [15, 16, 47], compatible with a double split-belt instrumented treadmill (Bertec, USA) provided by the German Aerospace Center (DLR) (see "*Study Hardware*" for treadmill specifications). The body suspension system

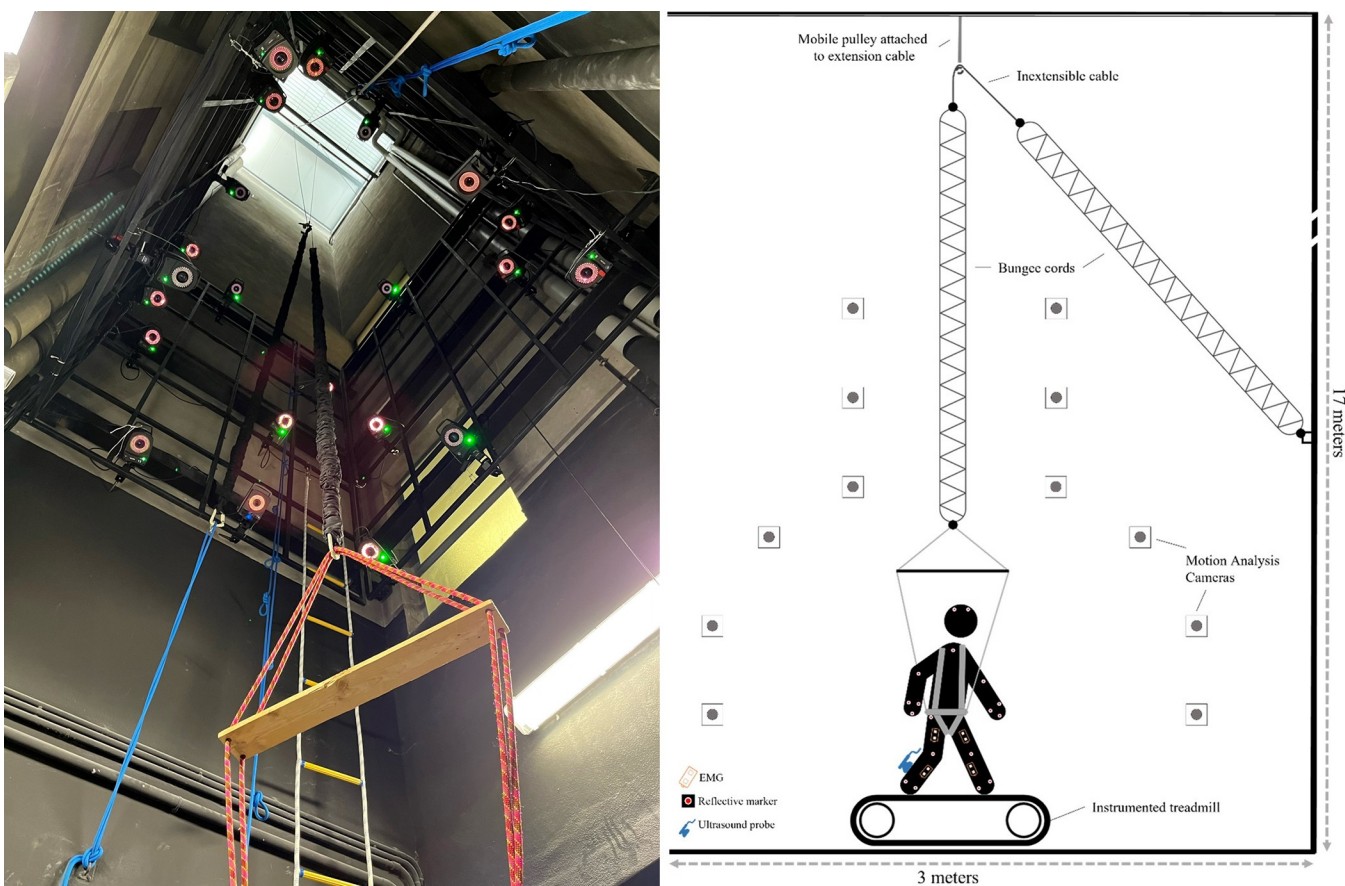

**Fig 2. L.O.O.P. facility.** *Left*: Picture of the cavaedium wherein the L.O.O.P has been installed, showing the two bungee cords, the inextensible short cable linking both bungee cords, and several of the motion analysis cameras. *Right*: Schematic of the experimental setup showing a participant in the harness connected to the bungee cords, walking on the treadmill in the middle of the capture environment and equipped with the EMG-sensors, reflective markers, and ultrasound probe.

consists of two bungee cords (Exploring Outdoor srl, Italy) with a resting length of 4 m and a stiffness of 92.7 N·m$^{-1}$. The bungee cords are linked in-series with an inextensible short cable (Dyneema SK78, diameter 4 mm, length 1.2 m, Gottifredi & Maffioli, Italy), working on an upper pulley. One of the bungee cords is fixed to the wall, while the other is connected to a force transducer (TS 300kg, AEP Transducers, Italy) placed in series with a body harness. The upper pulley can be raised or lowered by means of a suspension cable which is connected to a motorized winch (2.20 kW, Officine Iori SRL, Italy) allowing the titration (determined by the force transducer) of body (harness) unloading to the required gravity level (present study: 1, 0.7, 0.5, 0.38, 0.27, and 0.16g).

A key advantage of the L.O.O.P. compared to other hypogravity simulators results from the fact that the upper pulley is located high above the participant (~16 m), any horizontal forces generated due to fore-aft and/or lateral displacements during locomotion on the treadmill are minimised. For example, at 0.16g, a horizontal movement of 0.03 m with respect to the pulley results in an additional horizontal force of 0.92 N, representing just 0.4–0.7% of the peak push-off force during terrestrial (1g) stance [50]. Furthermore, the height of the shaft allows the use of a single pulley to accommodate the 20 m (when extended, 2 x 10 m) bungee cords limiting friction and displacement, independent of vertical force [15]. However, although high-fidelity simulated hypogravity is provided by generation of near constant vertical forces to the body's centre of mass (BCoM), swinging limbs remain subject to 1g –and thus can generate pendulum-like effects if exaggerated motion is performed. Thus, participants will be instructed to consider limb motion as stabiliser activity during jumping.

## Study hardware

**Treadmill.**   A double split-belt instrumented treadmill (total walking surface: 1.75 x 1m; speed: 0–5.83 m/s; Bertec, USA) will record 6-component loads from each belt (Fx, Fy, Fz, Mx, My, Mz) at a sampling rate of 2000 Hz which will be integrated into the 3D-Motion Capture System.

**3D-Motion analysis system.**   The L.O.O.P. is equipped with a 3D Motion Capture System (Vicon Motion Systems Ltd, UK) consisting of 24 cameras (Vicon MX Cameras, Vicon Motion System Ltd., UK) recording marker position at a frequency of 250 Hz, allowing full body segment tracking with a 66 marker set placed on anatomical landmarks according to the calibrated anatomical system technique [51] as described in Table 2 from which BCoM during locomotion, and the vertical displacement associated with maximal effort countermovement jumping are derived.

**Wireless surface electromyography.**   Sixteen wireless electromyography (EMG) sensors (Trigno, Delsys, USA) will be positioned according to SENIAM guidelines [52] to assess the bilateral myoelectrical activity of m. gluteus maximus, m. rectus femoris, m. vastus lateralis, m. biceps femoris (lateral hamstrings), m. semitendinosus (medial hamstrings), m. tibialis anterior, m. gastrocnemius and m. soleus. Prior to electrode placement each recording area will be shaved, abraded, and cleaned with an alcohol wipe/swab. Electrodes will be secured in place by medical tape (Leukoplast 2.5 cm x 5 m, BSN medical GmbH, Germany). EMG signals will be high-pass filtered, full-wave rectified, and low-pass filtered using a zero-lag second-order Butterworth filter. EMG amplitude will be normalised to each participant's maximum activation, defined as the maximum value recorded across the 1g dynamic trials for each muscle.

**Ultrasonography.**   Real-time B-mode ultrasound using a linear array transducer placed in a custom-made cast and secured with elastic bandages will be positioned over both the m. gastrocnemius medialis mid-belly to determine fascicle length and pennation angles [28, 53], and over the myotendinous junction (MTJ) to determine tendon length [54, 55]. The ultrasound recordings will be time-synchronized with the other data acquisition systems through a TTL

**Table 2. Description of marker locations used for the 3D-Motion analysis.**

| Segment | Static Markers | Dynamic Markers |
|---|---|---|
| *Pelvis* | Right Anterior Superior Iliac Spine<br>Left Anterior Superior Iliac Spine<br>Right Posterior Superior Iliac Spine<br>Left Posterior Superior Iliac Spine | Right Anterior Superior Iliac Spine<br>Left Anterior Superior Iliac Spine<br>Right Posterior Superior Iliac Spine<br>Left Posterior Superior Iliac Spine |
| *Right Femur* | Right Greater Trochanter<br>Right Lateral Femoral Epicondyle<br>Right Medial Femoral Epicondyle<br>Four-Marker Cluster | Right Lateral Femoral Epicondyle<br>Four-marker cluster |
| *Right Tibia-Fibular* | Right Lateral Malleolus<br>Right Medial Malleolus<br>Four-Marker Cluster | Right Lateral Malleolus<br>Four-Marker Cluster |
| *Right Foot Complex (Calcaneus, Talus, Toes)* | Right Posterior Calcaneus<br>Right Medial MTP5<br>Right Lateral MTP1<br>Right Superior MTP2 | Right Posterior Calcaneus<br>Right Medial MTP5<br>Right Lateral MTP1<br>Right Superior MTP2 |
| *Left Femur* | Left Greater Trochanter<br>Left Lateral Femoral Epicondyle<br>Left Medial Femoral Epicondyle<br>Four-Marker Cluster | Left Lateral Femoral Epicondyle<br>Four-marker cluster |
| *Left Tibia-Fibular* | Left Lateral Malleolus<br>Left Medial Malleolus<br>Four-Marker Cluster | Left Lateral Malleolus<br>Four-Marker Cluster |
| *Left Foot Complex (Calcaneus, Talus, Toes)* | Left Posterior Calcaneus<br>Left Medial MTP5<br>Left Lateral MTP1<br>Left Superior MTP2 | Left Posterior Calcaneus<br>Left Medial MTP5<br>Left Lateral MTP1<br>Left Superior MTP2 |
| *Torso* | Sternum<br>C7<br>Right Acromion Process<br>Left Acromion Process | Sternum<br>C7<br>Right Acromion Process<br>Left Acromion Process |
| *Right Humerus* | Right Lateral Humeral Epicondyle<br>Right Medial Humeral Epicondyle<br>Four-Marker Cluster | Right Lateral Humeral Epicondyle<br>Four-Marker Cluster |
| *Right Radius-Ulna* | Right Radial Styloid Process<br>Right Ulna Styloid Process<br>Four-Marker Cluster | Right Radial Styloid Process<br>Right Ulna Styloid Process<br>Four-Marker Cluster |
| *Left Humerus* | Left Lateral Humeral Epicondyle<br>Left Medial Humeral Epicondyle<br>Four-Marker Cluster | Left Lateral Humeral Epicondyle<br>Four-Marker Cluster |
| *Left Radius-Ulna* | Left Radial Styloid Process<br>Left Ulna Styloid Process<br>Four-Marker Cluster | Left Radial Styloid Process<br>Left Ulna Styloid Process<br>Four-Marker Cluster |
| *Hands* | No Markers (Locked Wrist Joint) | No Markers (Locked Wrist Joint) |

pulse passed through the auxiliary ECG channel. A semi-automated tracking algorithm (e.g., UltraTrack Software [56]) will be used to quantify the muscle fascicle length and pennation angles during the stance phase as previously used by our team [57, 58]. For the determination of tendon length, three markers are attached to the ultrasound transducer, enabling the transformation of the tracked fascicle insertion point close to the MTJ in the 2D ultrasound image into the global reference system. Additionally, the m. gastrocnemius medialis tendon length, fascicle length and pennation angle are determined while subjects are lying prone with the knee and ankle joints in anatomically neutral positions (knee fully extended, sole perpendicular to the tibia) and muscles are relaxed [53].

**Table 3. Description of the needed equipment and time for each part of the experimental sessions.**

| Activity—Equipment | Familiarisation ($T_0$) | Time (Hours) | Data Collection ($T_1$) | Time (Hours) |
|---|---|---|---|---|
| *Laboratory preparation* | | | | |
| Treadmill | X | 0.50 | X | 0.50 |
| L.O.O.P. Body Weight Support System | X | 0.25 | X | 0.25 |
| Ultrasonography System | | | X | 0.50 |
| EMG System | | | X | 0.25 |
| 3D Motion Capture System | | | X | 1.50 |
| *Participant Preparation* | | | | |
| Body Weight Support Harness | X | 0.50 | X | 0.25 |
| Ultrasonography Probe Placement | | | X | 0.50 |
| EMG Placement | | | X | 0.50 |
| Retroreflective Marker Placement | | | X | 0.50 |
| *Familiarisation Trials* | | | | |
| L.O.O.P. Body Weight Support System | X | 0.50 | | |
| *Calibration Trials* | | | | |
| Ultrasonography System | | | X | 0.25 |
| EMG System | | | X | 0.25 |
| 3D Motion Capture System | | | X | 0.25 |
| *Locomotion Trials* | | | | |
| Treadmill | | | X | 2.50 hours (Based on 1 min trial + 1 min rest + room for additional rest x 6 gravity levels) |
| L.O.O.P. Body Weight Support System | | | X | |
| Ultrasonography System | | | X | |
| EMG System | | | X | |
| 3D Motion Capture System | | | X | |
| *Plyometric Trials* | | | | |
| Treadmill | | | X | 2.50 hours (Based on 3 min trial + 3 min rest + room for additional rest x 6 gravity levels) |
| L.O.O.P. Body Weight Support System | | | X | |
| Ultrasonography System | | | X | |
| EMG System | | | X | |
| 3D Motion Capture System | | | X | |

Table 3 provides a comprehensive overview of all equipment used for each part of the experimental sessions.

## Outcome measures

**Anthropometrics.** For each participant, age (years), body mass (kg), -height (m) and leg length (m) will be collected (Table 4).

**Primary outcome measures–lower limb joint reaction forces.** Internal joint loading of the hip, knee and ankle (i.e., joint reaction forces, 'N') will be estimated by means of an optimal control framework (see *"Planned Data Analysis"*) using a direct collocation data-tracking method, derived from the collected experimental kinetic, kinematic, muscle activation and muscle-tendon unit behaviour data (*"Secondary Outcome Measures–Experimental Data"*, Table 4).

**Secondary outcome measures–experimental data.** *Kinetics.* Ground reaction forces ('N') extracted from the force instrumented treadmill will be collected, providing data on external forces acting on participants during movement.

**Table 4. Overview of the anthropometrics, primary and secondary outcome measures to be collected.**

| Outcome | | Unit | Study Hardware | Data Analysis Tool/ Software | Data Analysis Procedure |
|---|---|---|---|---|---|
| *Anthropometrics* | | | | | |
| Age | | Years | n.a. | n.a. | Manual |
| Body Mass | | Kilograms (kg) | Instrumented Treadmill | n.a. | Manual |
| Body Height | | Meter (m) | Stadiometer | n.a. | Manual |
| Leg Length | | Meter (m) | Tape measure | n.a. | Manual |
| *Primary Outcome Measures–Lower Limb Joint Reaction Forces* | | | | | |
| Hip Joint Reaction Forces | | Newton (N) | n.a. | Optimal Control Framework | Automated |
| Knee Joint Reaction Forces | | Newton (N) | n.a. | Optimal Control Framework | Automated |
| Ankle Joint Reaction Forces | | Newton (N) | n.a. | Optimal Control Framework | Automated |
| *Secondary Outcome Measures–Experimental Data* | | | | | |
| Kinetics | Ground Reaction Forces | Newton (N) | Instrumented Treadmill | Vicon Nexus | Automated |
| | Net Hip Joint Forces | Newton (N) | 3D-Motion Capture System & Instrumented Treadmill | Inverse Dynamics—OpenSim | Semi- Automated |
| | Net Knee Joint Forces | Newton (N) | 3D-Motion Capture System & Instrumented Treadmill | Inverse Dynamics—OpenSim | Semi- Automated |
| | Net Ankle Joint Forces | Newton (N) | 3D-Motion Capture System & Instrumented Treadmill | Inverse Dynamics—OpenSim | Semi- Automated |
| | Net Hip Joint Moment | Newton Meter (N·m) | 3D-Motion Capture System & Instrumented Treadmill | Inverse Dynamics—OpenSim | Semi- Automated |
| | Net Knee Joint Moment | Newton Meter (N·m) | 3D-Motion Capture System & Instrumented Treadmill | Inverse Dynamics—OpenSim | Semi- Automated |
| | Net Ankle Joint Moment | Newton Meter (N·m) | 3D-Motion Capture System & Instrumented Treadmill | Inverse Dynamics—OpenSim | Semi- Automated |
| Kinematics | Single Support Phase | Percentage gait cycle (%) | 3D-Motion Capture System & Instrumented Treadmill | Vicon Nexus | Semi- Automated |
| | Double Support Phase | Percentage gait cycle (%) | 3D-Motion Capture System & Instrumented Treadmill | Vicon Nexus | Semi- Automated |
| | Stride Frequency | Hertz (Hz) | 3D-Motion Capture System & Instrumented Treadmill | Vicon Nexus | Semi- Automated |
| | Stride Length | Meter (m) | 3D-Motion Capture System & Instrumented Treadmill | Vicon Nexus | Semi- Automated |
| | Flight Time* | Seconds (s) | 3D-Motion Capture System & Instrumented Treadmill | Vicon Nexus | Semi- Automated |
| | Contact Time* | Seconds (s) | 3D-Motion Capture System & Instrumented Treadmill | Vicon Nexus | Semi- Automated |
| | Hip Joint Angle | Degrees (°) | 3D-Motion Capture System & Instrumented Treadmill | Inverse Dynamics—OpenSim | Semi- Automated |
| | Knee Joint Angle | Degrees (°) | 3D-Motion Capture System & Instrumented Treadmill | Inverse Dynamics—OpenSim | Semi- Automated |
| | Ankle Joint Angle | Degrees (°) | 3D-Motion Capture System & Instrumented Treadmill | Inverse Dynamics—OpenSim | Semi- Automated |
| | Hip Joint Angular Velocity | Degrees per Second ($°·s^{-1}$) | 3D-Motion Capture System & Instrumented Treadmill | Inverse Dynamics—OpenSim | Semi- Automated |
| | Knee Joint Angular Velocity | Degrees per Second ($°·s^{-1}$) | 3D-Motion Capture System & Instrumented Treadmill | Inverse Dynamics—OpenSim | Semi- Automated |
| | Ankle Joint Angular Velocity | Degrees per Second ($°·s^{-1}$) | 3D-Motion Capture System & Instrumented Treadmill | Inverse Dynamics—OpenSim | Semi- Automated |
| | M. Gastrocnemius Medialis MTU Activation | n.a. | n.a. | Optimal Control Framework | Automated |

(*Continued*)

**Table 4.** (Continued)

| Outcome | | Unit | Study Hardware | Data Analysis Tool/ Software | Data Analysis Procedure |
|---|---|---|---|---|---|
| Muscle-Tendon Unit Geometries | M. Gastrocnemius Medialis Muscle Length | Millimetres (mm) | n.a. | Optimal Control Framework | Automated |
| | M. Gastrocnemius Medialis Pennation Angle | Degrees (°) | n.a. | Optimal Control Framework | Automated |
| | M. Gastrocnemius Medialis Tendon Length | Millimetres (mm) | n.a. | Optimal Control Framework | Automated |
| Muscle-Tendon Unit Forces | M. Gastrocnemius Medialis Muscle Force | Newton (N) | n.a. | Optimal Control Framework | Automated |
| | M. Gastrocnemius Medialis Tendon Force | Newton (N) | n.a. | Optimal Control Framework | Automated |
| Muscle Contraction Dynamics | M. Gastrocnemius Medialis Fascicle Length | Millimetres (mm) | Ultrasonography | MyoResearch software & UltraTrack Software | Semi-Automated |
| | M. Gastrocnemius Medialis Fascicle Velocity | Millimetres per Second (mm·s$^{-1}$) | Ultrasonography | MyoResearch software & UltraTrack Software | Semi-Automated |
| | M. Gastrocnemius Medialis Pennation Angle | Degrees (°) | Ultrasonography | MyoResearch software & UltraTrack Software | Semi-Automated |
| | M. Triceps Surae Tendon Strain | Millimetres (mm) | Ultrasonography | MyoResearch software & UltraTrack Software | Semi-Automated |
| Muscle Activation Dynamics | M. Gluteus Maximus Activity | Amplitude (mV) | Wireless Surface Electromyography | Delsys EMGworks Software | Automated |
| | M. Rectus Femoris Activity | Amplitude (mV) | Wireless Surface Electromyography | Delsys EMGworks Software | Automated |
| | M. Vastus Lateralis Activity | Amplitude (mV) | Wireless Surface Electromyography | Delsys EMGworks Software | Automated |
| | M. Biceps Femoris Activity | Amplitude (mV) | Wireless Surface Electromyography | Delsys EMGworks Software | Automated |
| | M. Semitendinosus Activity | Amplitude (mV) | Wireless Surface Electromyography | Delsys EMGworks Software | Automated |
| | M. Tibialis Anterior Activity | Amplitude (mV) | Wireless Surface Electromyography | Delsys EMGworks Software | Automated |
| | M. Gastrocnemius Activity | Amplitude (mV) | Wireless Surface Electromyography | Delsys EMGworks Software | Automated |
| | M. Soleus Activity | Amplitude (mV) | Wireless Surface Electromyography | Delsys EMGworks Software | Automated |

*Running, skipping and plyometric trials only.

Using OpenSim's [59, 60] inverse dynamics algorithms, net joint forces ('N') and net joint moments ('N·m') of the hip, knee and ankle will be determined for each dynamic trial.

*Kinematics.* Single and double support phases (percentage of the gait cycle, '%'), stride length ('m') and frequency ('Hz') of the locomotion trials will be determined, in addition flight and contact times (seconds, 's') will be collected for the running, skipping and plyometric trials.

Using OpenSim's [59, 60] inverse kinematics algorithms, the posture (i.e., positions and angles of the degrees of freedom, 'm' or '°') and velocities (i.e., the time derivative of the positions and angles, 'm·s$^{-1}$' or '°·s$^{-1}$', respectively) will be determined for each dynamic trial.

*Muscle-tendon unit behaviour.* Muscle-tendon unit behaviour of the m. gastrocnemius medialis will be modelled by determining the activation, geometries–muscle length (i.e., length of the contractile element, 'mm'), tendon length (i.e., length of the series elastic component, 'mm'), and pennation angle (i.e., the angle between the fascicle and the deep aponeurosis, '°')–, and forces–tendon force (i.e., force directed along the series elastic component, 'N'), muscle force (i.e., the summed forces from the contractile and parallel elastic components, 'N')–, using Hill-type muscle model formulations [61] (see *"Planned Data Analysis"*).

Additionally, muscle contraction dynamics of the m. gastrocnemius medialis will be determined through ultrasonography, measuring tendon strain (i.e., the change in distance between the myotendinous junction and the osteotendinous insertion represented by the midpoint of

the medial and lateral calcaneus markers, 'mm'), muscle fascicle length (i.e., the distance between the insertion of the fascicles into the superficial and the deep aponeuroses, 'mm'), pennation angle (i.e., the angle between the fascicle and the deep aponeurosis, '°'), and fascicle velocity (i.e., the time derivative of the fascicle length, 'mm·s$^{-1}$'). In addition, changes in fascicle length, changes in pennation angle, a modified version of the architectural gear ratio [62, 63], as well as m. gastrocnemius and m. soleus pre-activations [64] will be analysed.

Furthermore, muscle activation dynamics of m. gluteus maximus, m. rectus femoris, m. vastus lateralis, m. biceps femoris (lateral hamstrings), m. semitendinosus (medial hamstrings), m. tibialis anterior, m. gastrocnemius and m. soleus will be collected bilaterally.

## Planned data analysis

All collected experimental data will be used as input into an optimal control framework (Fig 3). This framework uses a direct collocation method to track experimental kinematics, net joint moments, and ground reaction forces for a single movement cycle. A movement cycle is defined as two successive contacts of the right foot (e.g., a stride) for the locomotion trials, and one contact and subsequent flight phase for the plyometric movements. The goal of the simulation is to minimise the cost function, consisting of a muscle-sharing term, data tracking terms and control variable minimisation terms, to estimate muscle activations for a given moment in time. The framework is thus designed to elicit a dynamically consistent simulation through tracking experimental data, whilst simulating joint reaction forces.

**The musculoskeletal model.** A generic OpenSim (Simbios, Stanford, California, USA [59, 60]) musculoskeletal model that has been validated for high knee flexion movement will be used within the framework (Fig 4, Lai *et al.* 2017 [65]). Skeletal motion within the framework is modelled with Newtonian rigid body mechanics and Hunt-Crossley foot-ground contacts. The model consists of 23-segments–ground, pelvis, torso, and, bilaterally, femur, patella, tibia-fibula, talus, calcaneus, toe, humerus, radius, ulna, and hand–with 37 degrees of freedom (DOF). Pelvis translation and rotation with respect to the ground will be modelled as a six DOF joint. The torso-pelvis, shoulder, and hip joints will be modelled as three DOF ball-and-socket joints, the wrists as two DOF universal joints, whilst ankle, subtalar, elbow, radioulnar and metatarsophalangeal (MTP) joints as single DOF hinge joints. The MTP and wrist joints will be locked at 0°. The tibiofemoral joint (knee) will be modelled as a single DOF hinge joint, with a 0–140° flexion range. The remaining tibia rotations and translations relative to the femur, and the sagittal plane patellofemoral joint motion (i.e., anteroposterior and vertical translation, and rotation about the mediolateral axis), will be defined by polynomials as a function of knee flexion.

The hip, knee, ankle, and subtalar DOF will be actuated by 80 Hill-type MTU, with the toes, torso and upper body driven by 19 ideal torque actuators. Idealised torques driving the non-muscle actuated DOF will be described as a function of activation and maximum torque. The foot-ground interactions will be modelled with six-spheres per foot–four attached to the calcaneus and two to the toe segments. Hunt-Crossley equations, modified to be double continuously differentiable [66], will be used to calculate the forces at each of the six spheres.

The generic musculoskeletal model will be linearly scaled within OpenSim. Dimension-specific scale factors per body segment will be determined by identifying the 3D distance between pairs of anatomical markers and by calculating the ratio between the experimental and modelled distances. These scale factors will be used to scale anthropometrics (i.e., segment length, width and depth), and inertial parameters (i.e., segment masses and moments of inertia) to each participant. A combination of automatic and manual scaling methods is used to scale muscle-tendon unit model parameters (see *"Muscle-Tendon Unit Modelling"*).

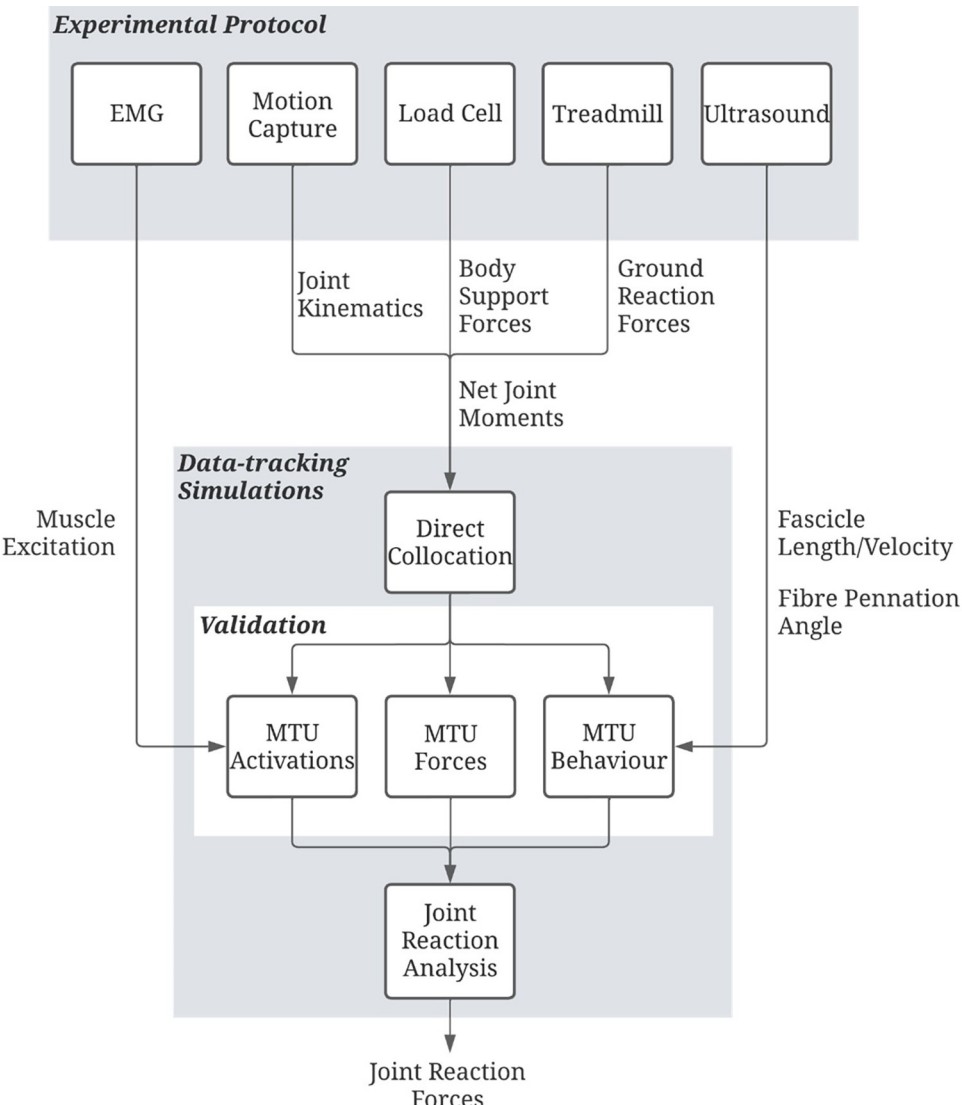

**Fig 3. A schematic of the data workflow to obtain the main outcome measure, joint reaction forces.** Experimental data are fed into the direct collocation optimal control framework to perform data-tracking simulations. Experimental EMG and ultrasound data are compared to simulated muscle-tendon unit (MTU) activations and behaviour, respectively, to validate the simulated MTU outcomes. The MTU outcomes are then used to inform a joint reaction analysis to calculate the joint reaction forces as a function of simulated kinematics, external loads (ground reaction and support forces), and MTU forces. EMG = electromyography.

*Muscle-tendon unit modelling*. Three-element Hill-type muscle model formulations will be used in this framework [61, 67, 68]. Briefly, the MTU complex consists of a contractile component, a passive elastic component parallel to the contractile component, and a series passive elastic component. Active (force-length and force-velocity) and passive (parallel and series) force generation are modelled via the dimensionless equations presented by De Groote *et al.*, [69]. The parameters maximum isometric force, optimum fibre length, pennation angle at optimum fibre length, maximum shortening velocity, and tendon slack length are used to describe the dimensionless formulations. The MTU parameters will be scaled based on the participant's anthropometrics (i.e., segment length, width and depth). Total MTU lengths will be adjusted during scaling such that the ratio of optimal fibre length to tendon slack length is maintained.

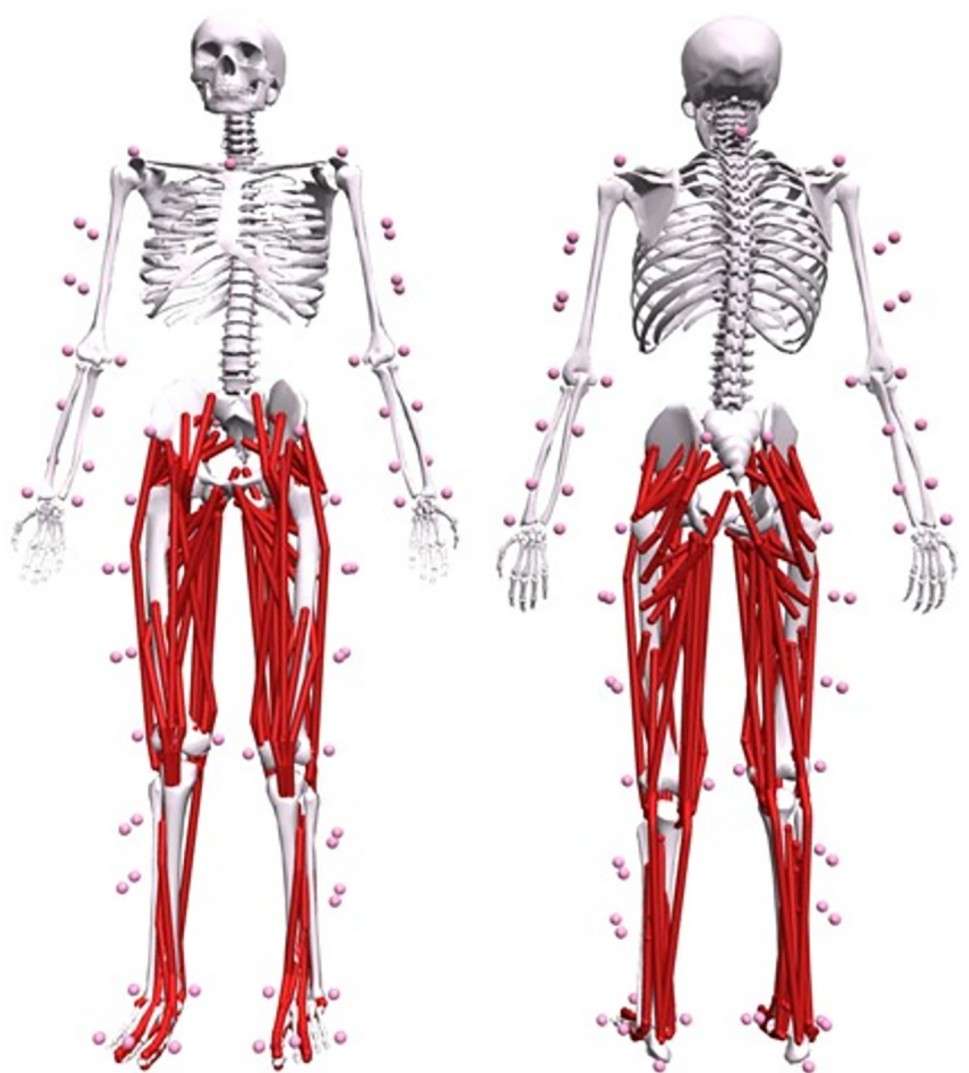

**Fig 4. The OpenSim musculoskeletal model that will be used for the analysis (Lai et al. 2017 [65]).** Red elements represent the muscle-tendon units, pink spheres represent marker placements on the participant.

Maximum isometric force will be updated by estimating physiological cross-sectional area from muscle volumes calculated as a function of participant height and mass [70] and the scaled optimal fibre lengths. Physiological cross-sectional area will then be multiplied by specific tension ($60 \, N/cm^2$), as performed previously [71]. Maximum shortening velocities are set to ten times the optimal fibre lengths [72]. Pennation angles at optimum fibre length from the unscaled model will be retained. Lengths, shortening velocities, and moment arms of MTUs will be defined as a function of joint positions and velocities [72, 73]. Polynomial coefficients will be determined for muscle lengths, velocities and moment arms, extracted using OpenSim's Muscle Analysis with the model positioned across a range of motion that exceeds the expected experimental data. Excitation-activation dynamics of the MTUs will be modelled via Raasch's activation model [74]–with modifications by De Groote *et al.*, [61].

**Optimal control framework.** The framework (Fig 3) will be formulated as optimal control problems (OCP) and will be implemented in MATLAB (Mathworks INC., USA) using CasADi (OPTEC, KU Leuven, Belgium [75]) and a modified version of OpenSim [59, 60] and

SimBody (Simbios, Stanford, California, USA [76]) to allow for algorithmic differentiation. Algorithmic differentiation (AD) allows for efficient and truncation-free evaluation of derivatives required by a non-linear program (NLP), which can lead to an almost 20-fold decrease in required simulation processing time [77].

Foot-ground contact sphere stiffness and damping (constant across all spheres) and their 3D position will be included as static parameters within the optimisation ($p_{cm}$). The remaining parameters (i.e., sphere radii, frictions, and transition velocity) will be kept constant. The state (x) and control (u) variables will be selected to allow efficient numerical formulation of the musculoskeletal system. The skeletal dynamics of the state variables, $q$ and $\dot{q}$, which correspond to the positions and velocities of each DOF, respectively, will be imposed by the DOF accelerations $\ddot{q}$. Muscle activations, $a_{MA}$, and normalised tendon forces, $F_t$, are introduced to describe the MTU state, with their first time derivatives, $\dot{a}_{MA}$ and $\dot{F}_t$, introduced as control variables to impose activation and contraction dynamics, respectively [69]. The states of idealised torque actuators will be described by their activations, $a_{TA}$, and controlled by their excitation, $e_{TA}$. Control variables will be introduced for the ground reaction forces, $u_{GRF}$, as performed previously [66]. This process will improve the convergence rate as the foot-ground contact sphere forces are subject to large fluctuations for small adjustments to the skeletal kinematics. Reserve actuators will be added to muscle-driven DOF as control variables, $u_{res}$, that describe the instantaneous moment being produced, to help convergence of the data tracking simulations.

The objective function will be formulated to minimise muscular effort and error between simulated and experimental data to promote physiologically realistic simulations. For instance, muscle-sharing (co-activation) will be achieved through minimisation of the summed muscle activations squared. Squared activations will be weighted by muscle volume to replicate a minimisation of muscular effort simulation [78]. This approach has been successfully used in submaximal [79] and maximal voluntary effort simulations [80] and is deemed appropriate for all the movements defined in our protocol. Data-tracking terms will be formulated as the squared error between experimental and simulated data for kinematics (angles and positions), ground reaction forces, and net joint moments. Tracking terms will be scaled to ensure terms are numerically similar within the objective function. Minimising the sum of squared terms will be included for reserve actuators and control variables ($\ddot{q}$, $\dot{a}_{MA}$, $\dot{F}_t$) [72, 80]. Each term will be weighted, determined via manual tuning, to achieve accurate data-tracking and physiologically realistic simulations. The validity of the simulations MTU activations and behaviour will be assessed through comparison with the experimental EMG and ultrasound data. The objective function, including term weightings, will be kept constant once calibrated.

Each OCP will be transcribed into NLP via a direct collocation method and solved using IPOPT [81]. Foot-ground contact model parameters will be included as static parameters, $p = [p_{CM}]$. Initially, the state ($x = [q, \dot{q}, a_{MA}, F_t, a_{TA}]$) and control ($u = [\ddot{q}, \dot{a}_{MA}, \dot{F}_t, u_{GRF}, e_{TA}]$) trajectories will be discretised across 50 equally spaced mesh points between the start and end of the movement cycle [78]. The number of mesh intervals will be reassessed and adjusted accordingly once data have been collected. The state trajectories will be further discretised into three-point intervals (collocation points) between mesh points using Legendre-Gauss-Radau quadrature and approximated with third-order polynomials. Net joint forces and moments via inverse dynamics, and Hunt-Crossley forces will be estimated at each mesh point as a function of the model's kinematics (i.e., $q$, $\dot{q}$, $\ddot{q}$), external forces (i.e., $u_{GRF}$ and body weight support forces), and foot-ground contact model parameters (i.e., $p_{cm}$). State, control and static variables will be bounded and then scaled within the NLP such that each variable falls within the interval -1 to 1 to improve the numerical conditioning of the NLPs [82].

A series of dynamic constraints will be imposed at each collocation point to maintain system dynamics. The skeletal dynamics and activation dynamics will be imposed as implicit constraints via first-order differential equations (e.g., $dq/dt - \dot{q} = 0$, $d\dot{q}/dt - \hat{A}q = 0$, $da_{MA}/dt - \dot{a}_{MA} = 0$, $dF_t/dt - \dot{F}_t = 0$). Explicit constraints will be imposed on the excitation-activation dynamics of the idealised torque actuators as the time delay between excitation and activation [72]. Path constraints are imposed at the beginning of each interval to achieve physiologically appropriate solutions. The muscle forces will be related to the experimental net joint moments via implicit constraints according to their polynomial-computed moment arms and the reserve actuators (i.e., $F_t$ times moment arm plus $u_{res}$). Additional constraints imposed dynamical consistency by setting pelvis residuals to zero. The $u_{GRF}$ controls will be matched to the foot-ground contact model as implicit constraints. Raasch's activation model will be imposed on the muscle activations via two inequality constraints based on the time constants for activation (0.015 s) and deactivation (0.06 s) of $\dot{a}_{MA}$ [61]. The Hill-equilibrium condition will be implicitly imposed by enforcing the muscle forces projected along the tendon to match the tendon forces. Continuality of the state variables between the end of collocation interval and the next time step will be enforced via implicit constraints. The cost function will be evaluated as the time integral between the start and end of the movement cycle, evaluated at each collocation point. The polynomials created to related joint kinematics (i.e., q and $\dot{q}$) to the MTU lengths, velocities, and moment arms will be called at each mesh point.

Best practise guidelines will be used after each simulation to assess tracking accuracy and physiological validity [83]. Tracking accuracy will be quantified via maximum and root mean squared error (RMSE) between simulated and experimental kinematics, GRFs and net joint moments. Simulated muscle activations will be qualitatively compared to the EMG in terms of timing and magnitude of the signals. Additionally, fascicle lengths and velocities, and pennation angles from the ultrasound images will be compared to the simulated MTU behaviour.

Once validated, simulated kinematics, GRFs and muscle-tendon unit forces will be used to calculate joint reaction forces using OpenSim's Analysis Tool [59, 60].

## Planned statistical analysis

Statistical significance will be assumed at $p < 0.05$. Normality of the data will be assessed via Kolmogorov-Smirnov testing complemented by visual inspection of QQ plots and histograms. Normally distributed data will be presented as mean (±SD), non-normally distributed data will be presented as median (±IQR).

The effect of high-fidelity simulated gravity (1, 0.7, 0.5, 0.38, 0.27 and 0.16g) upon lower limb internal kinetics (i.e., joint reaction forces) when walking, running, and skipping at speeds ranging between 0.53–3.6 m/s will be determined by a Two-Way Repeated Measures ANOVA with:

- Independent variables: six simulated gravity conditions; 10 locomotion trials (see Table 1)

- Dependent variables: Joint reaction forces of the hip, knee, and ankle

The effect of high-fidelity simulated gravity (1, 0.7, 0.5, 0.38, 0.27 and 0.16g) upon lower limb internal kinetics (i.e., joint reaction forces) during candidate plyometric movement (i.e., maximal ankle hopping, maximal effort countermovement jumping, and drop-jumps/landings) will be determined by a One-Way Repeated Measures ANOVA with:

- Independent variables: six simulated gravity conditions

- Dependent variables: Joint reaction forces of the hip, knee and ankle

In case the assumption of sphericity has been violated–i.e., Mauchly's test statistic is significant (p<0.05)–the Geisser-Greenhouse correction will be used to determine the effect of the independent variables on the dependent variables. If a significant effect of the simulated gravity condition, the locomotion trial, or the interaction between simulated gravity condition and locomotion trial is found, Bonferroni corrected post-hoc t-tests will be employed.

Alternatively, if data has a non-normal distribution the non-parametric Friedman test with Dunn's post-test will be used.

## Data management

Protection of all personal information collected throughout the experiment is ensured compliance with GDPR requirements:

- Data will be collected by the L.O.O.P. GBF personnel on an encrypted laptop and stored on a secure server, anonymized, backed up and then shared according to a data sharing agreement to the partners using an encrypted file sharing system. Therefore, all data files will be named with a key identifier that L.O.O.P. personnel will define and have responsibility to manage securely.

- The 'raw' kinematics, kinetics, and electromyography data will be saved in the Vicon 3D-Motion Capture System as.C3D files whilst ultrasound data will be saved in DICOM format. Both sets of files will be temporarily stored on encrypted data capture laptops. All files will be pseudonymised with a numeric code and kept separate from all other study data in a password protected folder.

- The 'raw' data will be uploaded on the same day of the data collection to a shared drive on a secure server by L.O.O.P. personnel. Access will be shared with the researchers from other institutions. A secured connection will be used to upload and download raw data from the shared drive. All academic partners will be able to access and download the data from this shared area and perform the analysis or calculation independently.

- As a number of institutions will be involved in the project, a data sharing agreement between institutions will be obtained but only pseudonymised data will be shared.

## Timeline

Depending on the current Covid restrictions, recruitment of participants will begin on November 1$^{st}$, 2022 and is expected to be completed by March 31$^{st}$, 2023.

## Summary

This study describes a protocol for the first time where a unique vertical body weight support system will be used to simulate, in high-fidelity, several locomotor and plyometric movements at a variety of gravity levels, including those resembling Lunar (0.16g) and Martian (0.38g) gravities. To our knowledge, this will be the first time such a facility has been used to simultaneously collect state-of-the-art movement kinetics, kinematics, muscle activation and muscle-tendon behaviour in a range of hypogravity conditions. Using these data as inputs within a computational musculoskeletal modelling and optimisation framework, we will estimate lower limbs internal kinetics (i.e., joint reaction forces) which is critical to gain a better understanding and prediction of hypogravity-induced musculoskeletal adaptations. Integration of muscle-tendon behaviour within the musculoskeletal modelling will significantly advance the

understanding of the importance of forces generated through muscular contractions for preserving musculoskeletal health in low gravity environments. Such insights on the relationship between movement in hypogravity and internal loads will be key to inform on future mitigation strategies and prevention of detrimental adaptations due to prolonged exposure to hypogravity but will also be crucial to guide Lunar surface extravehicular activity (EVA) operations, and EVA suit and habitat ergonomics.

## Conclusion

The results of the study will provide the first estimations of internal musculoskeletal loads associated with human movement performed in a range of hypogravity levels. This will be achieved through the collection of state-of-the-art kinetics, kinematics, muscle activation and muscle-tendon behaviour and its integration within a musculoskeletal modelling and optimisation framework. Thus, our unique data will be a key step towards modelling the musculoskeletal deconditioning associated with long term habitation on the Lunar surface, and thereby aiding the design of Lunar exercise countermeasures and mitigation strategies, while also guiding Lunar surface extravehicular (EVA) operations, and EVA suit and habitat ergonomics.

## Supporting information

**S1 File. Sample size calculations.**
(XLSX)

## Acknowledgments

The authors wish to thank the Cardiff Metropolitan University for providing additional equipment to complement the VICON 3D Motion Capture System.

## Author Contributions

**Conceptualization:** Nolan Herssens, James Cowburn, Kirsten Albracht, Bjoern Braunstein, Dario Cazzola, Alberto E. Minetti, Gaspare Pavei, Jörn Rittweger, Tobias Weber, David A. Green.

**Funding acquisition:** James Cowburn, Tobias Weber, David A. Green.

**Methodology:** Nolan Herssens, James Cowburn, Kirsten Albracht, Bjoern Braunstein, Dario Cazzola, Steffi Colyer, Alberto E. Minetti, Gaspare Pavei, Jörn Rittweger, Tobias Weber, David A. Green.

**Software:** James Cowburn.

**Supervision:** Nolan Herssens, Kirsten Albracht, Bjoern Braunstein, Dario Cazzola, Steffi Colyer, Alberto E. Minetti, Gaspare Pavei, Jörn Rittweger, Tobias Weber, David A. Green.

**Visualization:** Nolan Herssens.

**Writing – original draft:** Nolan Herssens, James Cowburn, Kirsten Albracht, Bjoern Braunstein, Dario Cazzola, Steffi Colyer, Alberto E. Minetti, Gaspare Pavei, Jörn Rittweger, Tobias Weber, David A. Green.

**Writing – review & editing:** Nolan Herssens, James Cowburn, Gaspare Pavei.

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
