## [Decision Letter · Decision Letter 0]

24 Oct 2022

PONE-D-22-19122Movement in Low gravity environments (MoLo) programme – The Molo-L.O.O.P. study protocolPLOS ONE

Dear Dr. Herssens,

I deeply apologiuze for the delay that characterized the processing of your paper. This was entirely due to the difficulty in securing available reviewers. Thank you for submitting your manuscript to PLOS ONE. After careful consideration, we feel that it has merit but does not fully meet PLOS ONE’s publication criteria as it currently stands. Therefore, we invite you to submit a revised version of the manuscript that addresses the points raised during the review process.

as you wil see the reviewer is enthusiastic about the work, but with a minor comment. Could you please return a manuscript with a sentence addressing the issue of skipping frequency and the suggested reference. Please not that, of course, the revised version would not go through review, but will undergo an instantaneous editroial evaluation. Please again accept my apologies for the difficulties in manuscript processing.

We look forward to receiving your revised manuscript.

Kind regards,

Luigi Cattaneo, MD, PhD

Academic Editor

PLOS ONE

Journal Requirements:

Reviewers' comments:

Reviewer's Responses to Questions

**Comments to the Author**

1. Does the manuscript provide a valid rationale for the proposed study, with clearly identified and justified research questions?

Reviewer #1: Yes

2. Is the protocol technically sound and planned in a manner that will lead to a meaningful outcome and allow testing the stated hypotheses?

Reviewer #1: Yes

3. Is the methodology feasible and described in sufficient detail to allow the work to be replicable?

Reviewer #1: Yes

4. Have the authors described where all data underlying the findings will be made available when the study is complete?

Reviewer #1: Yes

5. Is the manuscript presented in an intelligible fashion and written in standard English?

Reviewer #1: Yes

6. Review Comments to the Author

You may also provide optional suggestions and comments to authors that they might find helpful in planning their study.

Reviewer #1: The paper describes in details a research protocol to study locomotion and movements in low gravity environments. The manuscript is well written and the topic of great interests. The renewed interest in space exploration is providing new momentum to many research topics, including physio mechanics of locomotion.

The introduction is correctly focused on the topics of the investigation and the state of the art. References are sound. As one of the objective would be the estimation of joint reaction forces, maybe it would be worth to take into account the results of McDonnell et al. (2019, Gait & Posture 70, 414-419).

The protocol is very well designed. I only have one remark, or one question, about the range of speed and gaits. I noticed that the skipping trials would start at 1.94 m/s, and I'm wondering about the reasons of this choice. At lunar gravity the walk-run transition is estimated at a speed around 1 m/s, and it is known, from previous experiments and from the Apollo missions reports, that when astronauts can't walk, they start skip.

It is also true that the walk-skip gait change would imply an abrupt increase of the speed. However, I would have expected a wider range of speed for skipping trials.

7. PLOS authors have the option to publish the peer review history of their article (what does this mean?). If published, this will include your full peer review and any attached files.

Reviewer #1: **Yes: **Carlo M. Biancardi

---

## [Author Response · Author response to Decision Letter 0]

7 Nov 2022

Response to Reviewers

Editor Comments:

We have checked and made some changes to the style of the revised manuscript in order to adhere to the PLOS ONE style requirements. Please note that changes to the formatting have not been highlighted in the revised manuscript. 

We have checked both the “Funding information” and “Financial Disclosure” sections and both sections should now contain the same information. 

This work was supported by the European Space Agency: ESA contract No. 4000123348/18/NL/MH with the University of Bath; ESA contract DN RFP 3-16782/20/NL/AT MoLo Milano with the University of Applied Sciences Aachen.

The "Locomotion On Other Planets (L.O.O.P.): Hypogravity Analogue" of the University of Milan is supported by the European Space Agency as an ESA ground-based facility through the Continuously Open Research Announcement Opportunity for Ground-Based Facilities (ESA-CORA-GBF), ESA contract No. 4000137794/22/NL/PA/pt. 

This study was funded by CAMERA, the RCUK Centre of the Analysis of Motion, Entertainment Research and Applications, EP/M023281/1.

Authors DG and TW are employed by KBR GmbH on behalf of the European Space Agency. The funder KBR GmbH provided support in the form of salaries for the authors DG and TW but did not have any role in the study design, data collection, and analysis, decision to publish, or preparation of the manuscript. 

The funders had and will not have a role in study design, data collection and analysis, decision to publish, or preparation of the manuscript.

We have updated the Data Availability statement in the online submission environment to “Yes – any pilot data reported in this submission are fully available.” as the pilot data used to perform the sample size calculations are available as Supporting Information (S1 File).

We have also addressed this within the revised cover letter as requested.

We have reviewed the reference list and have made the following changes:

- We have added the following reference as suggested by Reviewer #1:

[40]: McDonnell J, Zwetsloot KA, Houmard J, DeVita P. Skipping has lower knee joint contact forces and higher metabolic cost compared to running. Gait Posture 2019;70:414–9. https://doi.org/10.1016/j.gaitpost.2019.03.028.

- We have removed the following references as the manuscript is still under preparation and is not essential to the figure it is pointing to, as the figure has been made specifically for the protocol submission, and is not included in the paper referred to:

Previously: [64] Cowburn J, Serrancolí G, Pavei G, Minetti A, Salo A, Colyer S, et al. The Biomechanical Handbook: a novel computational framework for the estimation of internal musculoskeletal loading in hypogravity. Manuscr Prep 2021.

Reviewer #1:

Reviewer #1: The paper describes in details a research protocol to study locomotion and movements in low gravity environments. The manuscript is well written and the topic of great interests. The renewed interest in space exploration is providing new momentum to many research topics, including physio mechanics of locomotion.

The introduction is correctly focused on the topics of the investigation and the state of the art. References are sound. As one of the objective would be the estimation of joint reaction forces, maybe it would be worth to take into account the results of McDonnell et al. (2019, Gait & Posture 70, 414-419).

Thank you for this comment, we have read the paper the reviewer has referred to and have added a statement discussing the relevant information within the introduction, lines 121-130.

The protocol is very well designed. I only have one remark, or one question, about the range of speed and gaits. I noticed that the skipping trials would start at 1.94 m/s, and I'm wondering about the reasons of this choice. At lunar gravity the walk-run transition is estimated at a speed around 1 m/s, and it is known, from previous experiments and from the Apollo missions reports, that when astronauts can't walk, they start skip.

It is also true that the walk-skip gait change would imply an abrupt increase of the speed. However, I would have expected a wider range of speed for skipping trials.

Thank you for this comment, we agree with the reviewer that at lunar gravity skipping will be adopted as the gait of choice at lower speeds than the two here analysed. We were forced to address skipping with just those two speeds due to a time restriction (the protocol would be longer of about twenty minutes) and also the knowledge that at 1, 0.7 and 0.5 g the skipping gait at 1.39 m/s is not that natural and meaningful. However, we will do our best to add the 1.39 m/s acquisition to Moon and Mars gravity, according to this right suggestion and in order to collect more interesting data.

---

## [Editor Report · Decision Letter 1]

9 Nov 2022

Movement in Low gravity environments (MoLo) programme – The MoLo-L.O.O.P. study protocol

PONE-D-22-19122R1

Dear Dr. Herssens,

We’re pleased to inform you that your manuscript has been judged scientifically suitable for publication and will be formally accepted for publication once it meets all outstanding technical requirements.

Kind regards,

Luigi Cattaneo, MD, PhD

Academic Editor

PLOS ONE
---

## [Editor Report · Acceptance letter]

14 Nov 2022

PONE-D-22-19122R1 

Movement in low gravity environments (MoLo) programme – The MoLo-L.O.O.P. study protocol 

Dear Dr. Herssens:

I'm pleased to inform you that your manuscript has been deemed suitable for publication in PLOS ONE. Congratulations! Your manuscript is now with our production department. 

Kind regards, 

on behalf of

Dr. Luigi Cattaneo 

Academic Editor

PLOS ONE